# Microdialysis Reveals Anti-Inflammatory Effects of Sulfated Glycosaminoglycanes in the Early Phase of Bone Healing

**DOI:** 10.3390/ijms24032077

**Published:** 2023-01-20

**Authors:** Sabine Schulze, Christin Neuber, Stephanie Möller, Jens Pietzsch, Klaus-Dieter Schaser, Stefan Rammelt

**Affiliations:** 1University Center for Orthopedics, Trauma and Plastic Surgery, University Hospital Carl Gustav Carus at TU Dresden, 01307 Dresden, Germany; 2Center for Translational Bone, Joint and Soft Tissue Research, Medical Faculty, TU Dresden, 01307 Dresden, Germany; 3Helmholtz-Zentrum Dresden-Rossendorf, Department of Radiopharmaceutical and Chemical Biology, Institute of Radiopharmaceutical Cancer Research, 01328 Dresden, Germany; 4Biomaterials Department, INNOVENT e. V., Prüssingstrasse 27B, 07745 Jena, Germany; 5Faculty of Chemistry and Food Chemistry, School of Science, Technische Universität Dresden, 01069 Dresden, Germany

**Keywords:** microdialysis, sulfated glycosaminoglycans, bone healing

## Abstract

Although chronic inflammation inhibits bone healing, the healing process is initiated by an inflammatory phase. In a well-tuned sequence of molecular events, pro-inflammatory cytokines are secreted to orchestrate the inflammation response to injury and the recruitment of progenitor cells. These events in turn activate the secretion of anti-inflammatory signaling molecules and attract cells and mediators that antagonize the inflammation and initiate the repair phase. Sulfated glycosaminoglycanes (sGAG) are known to interact with cytokines, chemokines and growth factors and, thus, alter the availability, duration and impact of those mediators on the local molecular level. sGAG-coated polycaprolactone-co-lactide (PCL) scaffolds were inserted into critical-size femur defects in adult male Wistar rats. The femur was stabilized with a plate, and the defect was filled with either sGAG-containing PCL scaffolds or autologous bone (positive control). Wound fluid samples obtained by microdialysis were characterized regarding alterations of cytokine concentrations over the first 24 h after surgery. The analyses revealed the inhibition of the pro-inflammatory cytokines IL-1β and MIP-2 in the sGAG-treated groups compared to the positive control. A simultaneous increase of IL-6 and TNF-α indicated advanced regenerative capacity of sGAG, suggesting their potential to improve bone healing.

## 1. Introduction

Bone healing is a complex process that occurs in three phases and involves numerous cells and cytokines orchestrating the recruitment, activation and inhibition of the participating players [1,2]. Due to the local trauma, a hematoma is generated at the fracture site, and the first, inflammatory phase is initialized by macrophages and neutrophils entering the site of injury. These cells release inflammatory cytokines such as interleukins (IL) 1β and 6 and TNF-α [3] that recruit progenitor cells such as mesenchymal stromal cells (MSC) and stimulate the differentiation and activation of these precursors to form vessels and rebuild the organic bone matrix. With that, the second, regenerative phase begins. Revascularization of the trauma site is essential for the invasion of precursor cells and mediators that initiate the synthesis of an organic scaffold, the soft callus, providing stability and serving as a framework for mineralization [3,4]. The original trabecular bone structure and shape are gradually restored within months after the trauma when the newly formed bone is rebuilt in the third, remodeling phase to adapt to the mechanical requirements and physiological loading [3,4,5].

Although bone regeneration is only completed with remodeling, complications within the initial inflammatory phase can have a major impact on the healing outcome. An uncontrolled inflammation causes delayed or impaired bone healing, as does the inhibition or suppression of the initial inflammation by NSAID, cortisone or other anti-inflammatory drugs [6].

New, adjuvant therapies to stimulate bone healing include artificial compounds that affect bone regeneration [7,8]. Artificial components of the natural extracellular matrix (ECM), namely, sulfated glycosaminoglycanes (sGAG), have gained increasing interest. sGAG, such as chondroitin sulfate (CS) and high-sulfated hyaluronan (sHA3), have been shown to have positive effects on bone healing [9]. These effects mainly rely on the capability of sGAG to alter the ECM of precursor cells [10], induce osteogenic differentiation in MSC and enhance mineralization [11]. These processes are thought to be promoted by the capability of sGAG to bind cytokines and growth factors and, with that, either enhancing their activity or shielding them from degradation [12,13]. However, these processes have not yet been monitored directly at the site of an acute bone defect. Microdialysis has evolved as a method of direct monitoring of the first stages of wound and bone healing by sampling directly at the site of injury [14,15].

In the present study, we analyzed the impact of sGAG on the decisive early inflammatory phase after a femoral osteotomy in rats. Therefore, biodegradable PCL scaffolds were coated with either collagen/sHA3 or collagen/CS and placed in critical-size femur defects. Treatment with autologous bone, the current gold standard for defect filling, served as a positive control. By applying microdialysis over the first 24 h after surgery, wound fluid was collected and analyzed regarding the concentrations of pro- and anti-inflammatory cytokines. We speculated that alterations in cytokine concentrations by sGAG may influence the mechanisms of bone–biomaterial interaction and the respective pathways, whose understanding has the potential to provide new insights leading to improved applications and therapies for bone defects.

## 2. Results

Twenty-three of twenty-four animals survived the surgery; one animal of the CS group died due to anesthesia complications. Another animal of the control group (12–24 h) died later during microdialysis and was observed and analyzed until 22 h after surgery. All animals subjected to microdialysis that were observed for a period between 12 and 24 h recovered well from the initial anesthesia and walked around freely in the cages between surgery and microdialysis. No allergic or other adverse reactions at the implantation site were noted over the whole observation period.

The concentration of IL-7 was below the detection limit in all dialysates.

In the positive control group receiving a bone graft (PC), IL-1β increased significantly 12 h after surgery compared to the level measured in the group receiving the sGAG treatment at any time interval (values from *p* = 0.034 (12–15 h control vs. 12–15 h sHA3) to *p* < 0.0001 (12–15 h control vs. 0–3 h CS and sHA3)) (Figure 1A). In animals with sGAG-coated scaffolds, IL-1β secretion was downregulated significantly compared to the level measured in the PC group (PC vs. sHA3 *p* = 0.021; PC vs. CS *p* = 0.009) (Figure 2A).

IL-6 concentration in the PC group increased within the 3–6 h interval and doubled 12–15 h after surgery, followed by a drop to the former concentrations (Figure 1B). The IL-6 secretion profile was altered by sGAG. In the CS group, an increased release started 3–6 h after surgery and remained high until the 9–12 h interval before it declined. The sHA3 group showed enhanced IL-6 secretion 9–12 h after surgery, which dropped 18 h after surgery. IL-6 secretion over 24 h was increased in animals treated with sGAG-coated scaffolds (Figure 2B).

The secretion of IL-10 did not vary significantly over time or with implant coating (Figure 1C). The analysis of secreted IL-10 over the total observation time of 24 h revealed decreased IL-10 secretion in the CS group; however, IL-10 concentration did not vary significantly compared to the PC and sHA3 groups (Figure 2C).

The release of MCP-1 increased until 12-15 h (PC) and 15–18 h (sHA3), respectively, followed by decreasing concentrations (Figure 1D). In the CS group, MCP-1 secretion was triggered as early as 3–6 h postoperatively, and peaked 6–9 h. The total MCP-1 concentration over 24 h increased in both, sHA3 and CS group compared to PC (Figure 2D).

MIP-2 secretion was induced in all groups from 12 to 24 h after surgery (Figure 1E). MIP-2 release increased earlier in the PC than in the sHA3- and CS-treated groups, and the secreted concentration was significantly increased in the PC group in the 12–15 h interval (compared to the CS group in the 12–15 h interval; *p* = 0.021). The secreted MIP-2 concentration over 24 h was decreased in both sGAG groups (Figure 2E).

TNF-α secretion showed a similar profile in the PC and sHA3 groups, with peaking concentrations 12–15 h post-operatively. In the CS group, TNF-α was significantly increased in the 3–6 h interval (*p* = 0.046) with an otherwise constant release over the observation period (Figure 1F). The animals in the sHA3 and CS groups secreted TNF-α in higher concentrations than those in the PC group over the observation time of 24 h (Figure 2F).

## 3. Discussion

The early events taking place at a fracture site and bone–implant interface are crucial for bone healing under physiological and pathological conditions. The effects of resorbable bone substitutes coated with artificial extracellular matrices containing collagen and sulfated glycosaminoglycans (sGAG) were for the first time observed during the first 24 h at the site of injury with microdialysis. The analyses revealed the inhibition of the secretion of the pro-inflammatory cytokines IL-1β and MIP-2 and the simultaneous increase in IL-6 and TNF-α in the sGAG-treated groups compared to the positive control.

IL-1β is a pro-inflammatory cytokine mainly produced by monocytes and macrophages. Its secretion is stimulated by tissue injury, among others. In the control group, the trauma-induced secretion started 12 h after osteotomy but was negligible in both sGAG groups. Together with the decreased IL-1β concentration, the data presented here indicate an anti-inflammatory effect of sGAG. Considering that IL-1β is known to stimulate its own expression in a positive feedback loop [16], sGAG appears to interfere with this feedback. Further possible explanations for the sGAG effect reported here are either the downregulation of IL-1β synthesis or the accelerated degradation of IL-1β.

Recent in vivo data revealed osteoclast activation and bone loss triggered by high IL-1β concentrations in inflammatory states [17]. In physiological conditions, IL-1β increases immediately after trauma and becomes undetectable within 72 h before its level rises again in the remodeling phase after several weeks [18]. In the present study, the presence of sGAG reduced IL-1β secretion within the first 24 h after osteotomy. Similarly, Förster et al. (2020) [9] reported supporting effects of sGAG on bone volume and stiffness at 12 weeks healing time in the identical defect model, confirming their role in enhancing bone healing. Additionally, decreased IL-1β concentrations improved vascularization and tissue maturation in soft tissue wounds [19]. The improved blood flow to newly formed tissue might also enhance bone healing.

IL-1β triggers the synthesis of the acute-phase mediator IL-6 [16]. The increase in IL-1β and IL-6 12-15 h after osteotomy in the positive control confirmed the IL-1β-dependent induction of IL-6 expression, taking into account that the intervals of collection covered three hours. In the sHA3 and CS groups, however, IL-6 secretion onset occurred in the 3-6 h (CS) and 9-12 h intervals (sHA3), suggesting an activation of IL-6 synthesis that was independent of IL-1β. A previously published study on microdialysis in bone defects showed an IL-6 secretion profile similar to the one we found in the control group [15]. These authors described increased IL-6 concentrations peaking 12–15 h after surgery in an empty critical-size femur defect. In the present experiment, a similar effect was found in the positive control treated with autologous bone. In contrast, the treatment with CS and sHA3 resulted in an earlier onset of IL-6 secretion and increased overall IL-6 concentrations. This early, prolonged and enhanced immune response compared to the control group may be attributed to the addition of sGAG. Together with its promotion of osteogenic differentiation of precursor cells and the induction of mineralization [17,18], the anti-inflammatory effect of IL-6 might contribute to the enhanced healing capacity reported for CS in in vivo studies [9,20], and the altered cytokine secretion might be associated with CS-mediated improved bone healing and defect regeneration [5,21].

IL-10 is produced by monocytes and macrophages, has inhibiting properties in inflammatory states and suppresses the immune response [22]. This anti-inflammatory cytokine promotes MSC migration and osteogenesis during bone regeneration [17]. A downregulation of IL-6 or TNF-α by IL-10, as described previously by Kessler and collaborators [22], could not be observed in the present experiments. A potential reason for this could be the length of the studied period, which was 48 h in the study by Kessler et al. [22], while our study covered the first 24 h after surgery. The measured IL-10 concentrations did not vary significantly between the implant groups or over the observation time. IL-10 is essential for T-cell induction, but high levels shortly after trauma are reported to compromise bone regeneration [23]. The observed minor fluctuations in IL-10 concentration within the first 24 h after surgery indicated a physiological healing process in all groups.

MCP-1 is a pro-inflammatory cytokine which is released by endothelial cells and macrophages as well as by osteoblasts and osteoclasts following bone injuries [6]. Additionally, MCP-1 is secreted by neutrophils within the fracture hematoma for the recruitment and tissue infiltration of monocytes and macrophages [24,25,26]. Finally, MCP-1 orchestrates the migration of mesenchymal stromal cells (MSC), the progenitor cells of osteoblasts [17], and recruits hematopoietic stem cells. It thus plays an essential role in the onset of bone healing [27]. By the interruption of the MCP-1 signaling pathway, macrophage migration is inhibited, and thus, bone regeneration is impaired, demonstrating the essential need for inflammation in the early healing phases. However, uncontrolled MCP-1 secretion, as it occurs in pathological inflammatory states, causes persistent inflammation that impairs the transition to the repair phase of healing [19]. On the other hand, a lack of MCP-1 has drawbacks such as dysfunctional vascularization, leading to delayed or impaired bone healing [28] due to reduced macrophage infiltration, delayed degradation of cell debris and inhibited mineralization [29].

Wang et al. (2015) [30] found that MCP-1 activates the expression of IL-6. When comparing the secretion profiles of MCP-1 and IL-6 detected here, the data indicated a correlated induction of both cytokines, confirming the findings of Wang et al. The concentration of secreted MCP-1 was highest in the CS group, and the secretion increased earlier than in the PC and sHA3 groups. As MCP-1 promotes early macrophage infiltration finally resulting in improved bone healing [24], our results indicate a supportive effect of CS on bone regeneration. Whether the fast downregulation of MCP-1 in the CS group potentially leads to a shorter time to bone healing requires further analyses.

The pro-inflammatory cytokine MIP-2 is released by macrophages and monocytes in response to trauma or infection [31] to recruit granulocytes in the early healing phase [32]. Diab et al. (1999) [33] investigated MIP-2 expression in rats during bacterial meningitis and found an increase from 12 to 48 h after infection. This correlates with the enhanced MIP-2 concentrations we found 12-24 h after surgery at the defect site. In the sGAG groups, MIP-2 secretion started 3-6 h (CS) and 9-12 h (sHA3) after surgery, suggesting an early onset of the immune response. The overall concentration of MIP-2 in the sGAG groups lagged behind that of the control group, supporting the hypothesis of an immunosuppressive effect of sGAG, as seen for IL-1β.

The pro-inflammatory cytokine TNF-α is secreted by macrophages in inflammatory tissue states in order to recruit immune cells and progenitor cells [28]. Depending on its concentration and the duration of TNF-α secretion, TNF-α can either inhibit or promote bone formation [17]. The short-term release of TNF-α recruits precursor cells, induces osteogenic differentiation in MSC in vitro, increases mineralization [29] and essentially promotes bone regeneration in vivo [17]. A prolonged secretion as well as high concentrations of TNF-α, however, downregulate the osteogenic differentiation of osteoblast precursors [28], inhibit collagen synthesis [34] and mineralization and can induce apoptosis in MSC as well as in mature osteoblasts [17], leading to reduced bone volume and impaired bone regeneration. Additionally, in chronic inflammation, TNF-α induces the secretion of M-CSF which stimulates osteoclastogenesis [17]. Although a significantly higher concentration of TNF-α was observed in the CS group 3-6 h after surgery, the overall release of TNF-α was higher in the sHA3 group.

However, to identify states of unregulated, chronic inflammation in the animals, longitudinal repeated measurements beyond the initial inflammatory phase would be required. Methods such as positron emission tomography (PET) or optical imaging (OI) facilitate longitudinal observations within individual animals, allowing not only the identification of inflammatory states but also the functional characterization of the molecules involved in these processes [2,20].

There are a few limitations to this study. As we decided to split the observation time into halves (0–12 h and 12–24 h), in order to reduce stress for the animals due to an excessively long anesthesia, inter-individual variances between the animals regarding the secreted cytokine concentrations could have affected and enhanced differences in the release profiles. Additionally, the fact that the animals included in the group subjected to 12–24 h of observation woke up from the anesthesia shortly after surgery and moved in their cages could also have influenced the secretion of various cytokines.

## 4. Materials and Methods

### 4.1. sGAG Synthesis

High-molecular weight hyaluronan (HA, isolated from *Streptococcus*, M_w_ = 1,200,000 g mol^−1^ kDa) was purchased from Aqua Biochem (Dessau, Germany). Chondroitin sulfate (CS, from bovine trachea, a mixture of 70% chondroitin-4-sulfate and 30% chondroitin-6-sulfate, M_w_ 21.6 g/mol) was also purchased from Kraeber GmbH and purified by dialysis. The high-sulfated HA (sHA3) based on high molecular HA was synthesized as described [35,36]. For the preparation of sHA3, we used a ratio of OH/SO_3_ = 1:20 with one hour reaction time. SO_3_/dimethylformamide was used as the sulfating reagent. The derivatives were characterized by elemental analysis and nuclear magnetic resonance (NMR); the determination of the molecular weight was performed by gel permeation chromatography (GPC) with a triple detection system consisting of a laser light scattering (LLS) detector, a refractive index (RI) detector and a UV-detector. The absolute values of number-average (M_n_) and weight-average (M_w_) molecular weights were determined using the LLS detection system. The calculation of polydispersity (PD = M_w_/M_n_) was performed on the basis of the M_n_ and M_w_ values obtained from RI detection [37,38].

The chemical structures and characteristics of the respective GAG derivatives are provided in Figure 3 and Table 1.

### 4.2. Preparation and Coating of the PCL Scaffolds

The coating of the PCL scaffolds with sGAG was performed as describe elsewhere [9,20]. In brief, polycaprolactone-co-lactide (PCL) scaffolds were prepared as previously described [9,39]. Resorbable PCL monofilament fibers, size 0.7 (6-0), produced by Gunze Ltd. (Osaka, Japan), were purchased from Catgut (Markneukirchen, Germany) and embroidered on polyvinyl alcohol, obtaining scaffolds 5 mm in diameter (Rahmig & Partner Embroidery, Ellefeld, Germany). The scaffolds were removed from the ground fabric by repeated washing with deionized water (5 × 20 min) and n-heptane (3 × 10 min). The scaffolds were hydrophilized in 1 M NaOH in 50% methanol for 15 min. After washing with deionized water, the scaffolds were air-dried for further processing.

For PCL coating, collagen (col)/sGAG solutions were prepared by mixing equal volumes of col and sGAG derivatives with a molarity of 5 mM disaccharide units each. The PCL scaffolds were incubated in the respective coating solutions for 16–18 h at 37 °C. The scaffolds were removed from the solutions, frozen for 2 h at −80 °C and freeze-dried overnight. After two washing steps with deionized water, the scaffolds were air-dried under laminar flow for 30 min, frozen for 2 h at −80 °C and freeze-dried again.

Five coated scaffolds were piled up on top of each other and fixed with a resorbable PCL suture into stacks of 5 mm of height according to the defect size. The scaffolds were sterilized by using gamma irradiation at 15 kGy (BBF Sterilisationsservice, Kernen, Germany).

### 4.3. Animal Surgery

The study was approved by the local Institutional Animal Care Committee and the local authorities (DD24-9168.11/1/421). At 12 weeks of age, 24 male Wistar rats were obtained from Charles River Laboratories (Wilmington, MA, USA) and were randomized to three groups: (1) porcine gelatin sponge (Spongostan^®^, Ethicon, Bridgewater, NJ, USA) mixed with bone chips from the resected part of the femur (positive control, PC), (2) PCL coated with collagen and chondroitin sulfate (CS), (3) PCL coated with collagen and hypersulfated hyaluronic acid (sHA3). The animals were housed in a light/dark cycle of 12:12 h at a constant temperature, with access to food and water ad libitum.

The surgical procedure was performed as described by Neuber et al. (2019) [20] and Förster et al. (2020) [9]. Under general anesthesia (intraperitoneal injection of 100 mg/kg of body weight of ketamine and 10 mg/kg of body weight of xylazine), the right hind leg was shaved, and a 3 cm incision was made to expose the femur. For internal fixation, a 5-hole titanium plate was fixed with 4 screws, leaving the central screw hole blank (Figure 4A). A critical 5 mm cross-sectional mid-shaft defect was created using a wire saw (Figure 4B). The scaffolds or bone chips were inserted press-fit into the bone defect (Figure 4C). The muscle was sewed using absorbable suture, and the skin was closed with skin clips. All animals received Carprofen (5 mg/kg of body weight) subcutaneously for general analgesia after surgery. In order to reduce the stress levels in the animals, the observation time of 24 h was split into two 12 h intervals. The animals in the 0–12 h-interval group remained under anesthesia after surgery. The other group was observed for 12-24 h post-surgery. These animals recovered from anesthesia, were allowed to move freely and received anesthesia again after 12 h. To avoid dehydration and hypoglycemia during microdialysis, all animals received 0.25 mL of 0.9% NaCl and 0.25 mL 10% of glucose every 120 min. The animals were sacrificed at the end of the experiment.

### 4.4. Microdialysis

The procedure followed the descriptions published by Förster et al. (2016) [14,15]. Perfusion fluid was prepared (147 mmol/L NaCl, 4 mmol/L KCl, 2.3 mmol/L CalCl_2_, 1% bovine serum albumin (Sigma, Taufkirchen, Germany)) and pumped through a catheter for priming. Using a CMA 402 microdialysis pump (CMA Microdialysis, Kista, Sweden), a flow rate of 2 µL/min was applied. When the microdialysis catheter was inserted in the bone defect/implant, the dialysates were collected on ice for 12 h in 3 h intervals. Each collection tube contained the protease inhibitor phenylmethylsulfonylfluoride (PMSF, 1 mmol/L) to avoid cytokine degradation. The dialysates were stored at −20 °C until analysis.

### 4.5. Multiplex ELISA

The cytokine concentrations were determined by a bead-based multiplex immunoassay according to the manufacturer’s instructions. The Bio-Plex Pro Rat Cytokine Assays (Bio-Rad, München, Germany) for interleukin (IL)-1β, IL-6, IL-7 IL-10, monocyte chemoattractant protein (MCP-)1, macrophage inflammatory protein (MIP-)2 and tumor necrosis factor (TNF-)α were performed with the Bio-Plex 200 system using Bio-Plex Manager Software 6.1 (Bio-Rad, München, Germany).

### 4.6. Statistical Analyses

All multiplex ELISAs were performed in duplicates. The data are presented as mean ± SEM. Statistical analyses was carried out using ANOVA with Tukey post-hoc test. *p* values < 0.05 were considered to be statistically significant.

## 5. Conclusions

By using microdialysis, wound fluid and fracture hematoma were collected within the first 24 h after creation of a bone defect in order to identify and quantify acute-phase mediators and to characterize the impact of sGAG on cytokine secretion. The concentration of the pro-inflammatory IL-1β and MIP-2 was substantially decreased by sGAG. In contrast, the levels of the cytokines IL-6 and TNF-α, which are reported to recruit progenitors, enhance vascularization and induce osteogenesis, were increased in the sGAG groups. These results suggest a positive effect of sGAG on bone healing, which is in agreement with previous in vivo and in vitro data. Our investigation focused on the first stages of bone healing in order to gain insight in the decisive early processes at the site of an acute bone defect (Figure 5). Tracking the subsequent healing processes beyond the first 24 h will require further studies with alternative techniques providing individual longitudinal monitoring.

## Figures and Tables

**Figure 1 ijms-24-02077-f001:**
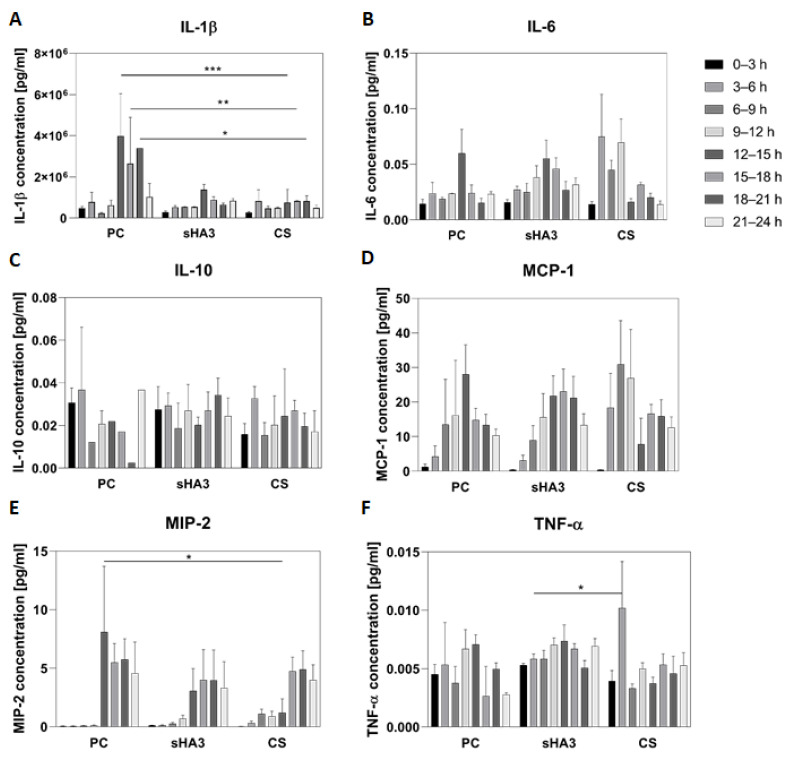
Cytokine concentrations in the wound fluid within a critical-size femoral defect after osteotomy in rats. The defect was filled with either spongostan and autologous bone (positive control, PC) or a PCL scaffold containing collagen and hypersulfated hyaluronic acid (sHA3) or chondroitin sulfate (CS). After osteotomy, the wound fluid was collected for 24 h in intervals of 3 h, using microdialysis, and the concentration of the cytokines IL-1β (**A**), IL-6 (**B**), IL-10 (**C**), MCP-1 (**D**), MIP-2 (**E**) and TNF-α (**F**) was determined by a multiplex immunoassay. Data are presented as mean ± SEM, statistical significance is indicated with * (*p* < 0.05), ** (*p* < 0.01), *** (*p* < 0.005).

**Figure 2 ijms-24-02077-f002:**
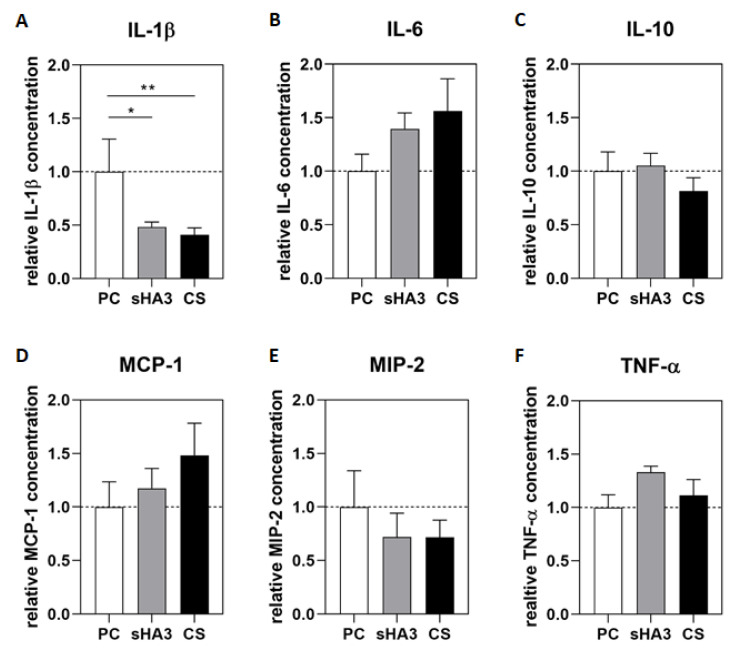
Cytokine concentrations over 24 h of collection time. After osteotomy, the wound fluid was collected for 24 h, and the concentration of the cytokines IL-1β (**A**), IL-6 (**B**), IL-10 (**C**), MCP-1 (**D**), MIP-2 (**E**) and TNF-α (**F**) was determined by a multiplex immunoassay. Data are presented as mean ± SEM relative to PC (mean of the PC represented by dotted line), statistical significance is indicated with * (*p* < 0.05), ** (*p* < 0.01).

**Figure 3 ijms-24-02077-f003:**
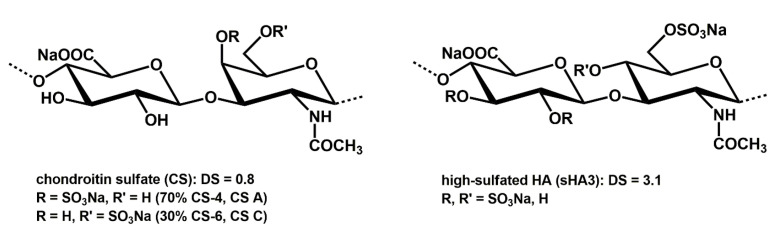
Structural characteristics of the sulfated GAG derivatives.

**Figure 4 ijms-24-02077-f004:**
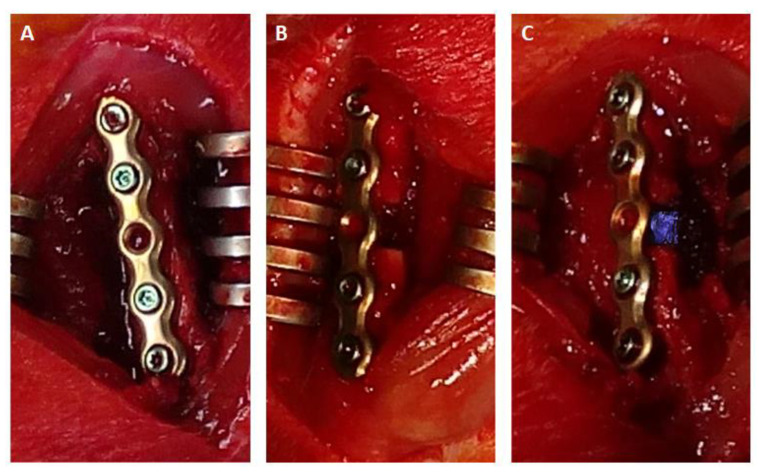
Surgical procedure. (**A**) Fixation of the right femur with a five-hole titanium plate and four screws. (**B**) A 5 mm critical-size defect was created. (**C**) Press-fit insertion of a PCL scaffold into the bone defect.

**Figure 5 ijms-24-02077-f005:**
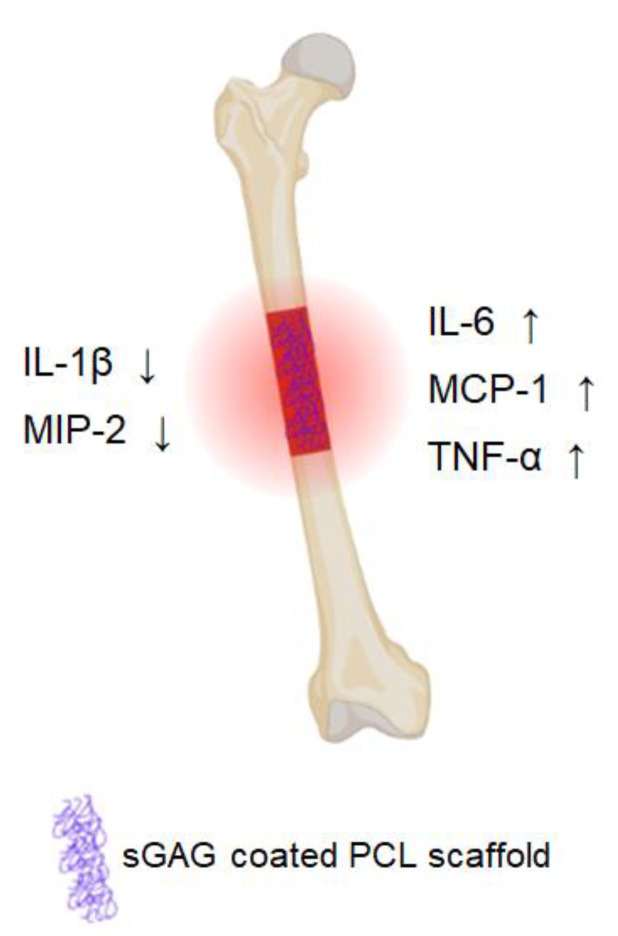
Impact of sulfated glycosaminoglycans (sGAG) on the local concentrations of cytokines released in the first 24 h after trauma. The concentration of the proinflammatory cytokines IL-1β and MIP-2 decreases when a critical-size femoral defect in rats was grafted with a sGAG-coated PCL scaffold. On the other hand, more IL-6, MCP-1 and TNF-α were secreted, which attenuated the inflammatory response and initiated regeneration processes.

**Table 1 ijms-24-02077-t001:** Analytical data of the synthesized GAG derivatives.

Sample	CS	sHA3
DS	0.8	3.1
M_n_ [g mol^−1^]	18,653 (38,540)	34,753 (46,845)
M_w_ [g mol^−1^]	21,592 (60,434)	51,268 (80,810)
PD	1.6	1.7

CS: purified chondroitin sulfate; sHA3: high-sulfated HA, DS: average number of sulfate groups per repeating disaccharide unit; M_n_: number-average molecular weight, M_w_: weight-average molecular weight, analyzed by gel permeation chromatography (GPC); values as determined with Laser Light Scattering detection and Refraction index (RI) detection (in parentheses); PD: polydispersity index detected by GPC; values calculated from RI detection.

## Data Availability

The full datasets generated for this study are available from the corresponding author on request.

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
