# Peer review of "Microdialysis Reveals Anti-Inflammatory Effects of Sulfated Glycosaminoglycanes in the Early Phase of Bone Healing"

_ijms, 2023, doi:10.3390/ijms24032077_

Round 1

Reviewer 1 Report

Schulze et al. analyze in this study the impact of scaffolds consisting of sulfated glycosaminoglucanes (sGAG) coated to polycaprolactone (PLC) in a rat model of bone healing. They found that wound fluids of rats treated with sGAG-scaffolds contained less IL-1beta and MIP-2. The wound-fluids were obtained by microdialysis and the cytokines were analyzed by standard multiplex ELISA. The application of anti-inflammatory scaffolds in tissue repair and wound healing is an emerging field and the results of such in vivo studies could be certainly important for the field. Yet, key issues with the current version of the paper preclude publication in my opinion.

1.     The authors provide no data/analyses about the biochemical and physical properties of the scaffolds used in this study.

2.     Photos and clinical data of the animal surgery should be presented in the paper.

3.     Figure 1, the amounts/concentrations of the measured cytokines are difficult to understand and show unusual levels. For instance, according to the labelling of the figure, IL-1 seems to be present in the microgram range (which is extremely high!), whereas other cytokines are (far) below 1 pg/ml in the femto-gram range. I am not aware of an ELISA system measuring cytokine concentrations below 1 pg/ml. 

4.     The results section is too “minimalistic” for a scientific paper and the story is actually based only on 1 figure. I assume that the authors have analyzed and checked a number of other factors or clinical parameters which could make the study more vivid and convincing.

Author Response

  1. The authors provide no data/analyses about the biochemical and physical properties of the scaffolds used in this study.

Our group has been working with PCL for several years and has published several papers showing properties such as pore size (Rentsch, C., Rentsch, B., Heinemann, S., Bernhardt, R., Bischoff, B., Förster, Y., ... & Rammelt, S. (2014). ECM inspired coating of embroidered 3D scaffolds enhances calvaria bone regeneration. BioMed Research International, 2014) and stiffness (Förster, Y., Schulze, S., Penk, A., Neuber, C., Möller, S., Hintze, V., ... & Rammelt, S. (2020). The influence of different artificial extracellular matrix implant coatings on the regeneration of a critical size femur defect in rats. Materials Science and Engineering: C, 116, 111157) of the woven scaffolds. These references are included in the manuscript.

  1. Photos and clinical data of the animal surgery should be presented in the paper.

Thank you for your interest in surgical and clinical details. We have added a photograph of the experimental setup and the surgical site. The method itself has been described in more detail in Neuber, C., Schulze, S., Förster, Y., Hofheinz, F., Wodke, J., Möller, S., ... & Pietzsch, J. (2019). Biomaterials in repairing rat femoral defects: In vivo insights from small animal positron emission tomography/computed tomography (PET/CT) studies. Clinical Hemorheology and Microcirculation, 73(1), 177-194. We refer to this publication in the revised manuscript. Additional photos can be found in Förster, Y., Gao, W., Demmrich, A., Hempel, U., Hofbauer, L. C., & Rammelt, S. (2013). Monitoring of the first stages of bone healing with microdialysis. Acta Orthopaedica, 84(1), 76-81, that is also referenced.

We have added another statement on the clinical course during the experiments. Clinical details such as wound and bone healing, weight-bearing and final bone strength were beyond the scope of this study as the experiments were terminal due to the long duration of anesthesia. However, we have published the impact of sulfated glycosaminoglycans on bone regeneration over a longer period in the same defect model in Förster, Y., Schulze, S., Penk, A., Neuber, C., Möller, S., Hintze, V., ... & Rammelt, S. (2020). The influence of different artificial extracellular matrix implant coatings on the regeneration of a critical size femur defect in rats. Materials Science and Engineering: C, 116, 111157. This study is referenced in the present manuscript.

  1. Figure 1, the amounts/concentrations of the measured cytokines are difficult to understand and show unusual levels. For instance, according to the labelling of the figure, IL-1 seems to be present in the microgram range (which is extremely high!), whereas other cytokines are (far) below 1 pg/ml in the femto-gram range. I am not aware of an ELISA system measuring cytokine concentrations below 1 pg/ml.

These issues came into our minds, too, and we addressed them with repeated measurements and expert assistance but the results remained the same. The reproducible variation over time for various cytokines (i.e. IL-6 and TNF-α) indicates robust results.

  1. The results section is too “minimalistic” for a scientific paper and the story is actually based only on 1 figure. I assume that the authors have analyzed and checked a number of other factors or clinical parameters which could make the study more vivid and convincing.

The discussion is focused and based on extensile information from two figures in the manuscript. As mentioned above, in this study we specifically observed the early healing phase and analyzed the impact of sulfated glycosaminoglycans on the cytokine release within the first 24 hours after trauma. To the best of our knowledge, this has not been reported before. The microdialysis was planned as terminal experiment and officially approved by the animal care committee. Therefore, we cannot report on more clinical details than those mentioned in the revised manuscript. For further information on long time healing results we refer to our previously published studies using the same animal model but with substantially shorter time under anesthesia. The goal of the present study was to reveal the underlying mechanisms that take place in the decisive first hours after implantation of the scaffolds. This is pointed out more clearly in the revised manuscript (see also comments to reviewer 3).

Reviewer 2 Report

The authors analyze the impact of sGAG on the early inflammatory phase  after osteotomy in rats. Therefore, biodegradable PCL scaffolds were coated with either  collagen/sHA3 or collagen/CS and placed in critical size femur defects. Treatment with autologous bone, the current gold standard for defect filling, served as positive control.  By applying microdialysis over the first 24 h after surgery wound fluid was collected and  analyzed regarding concentrations of pro- and anti-inflammatory cytokines. Alterations  in cytokine concentrations by sGAG may characterize the mechanisms of bone biomaterials interaction and the respective pathways potentially leading to improved applications  and therapies of bone defects. In conclusion, by using microdialysis wound fluid and fracture hematoma was collected to identify and quantify acute phase mediators within the first 24 h after trauma and to characterize the impact of sGAG on the cytokine secretion. The concentration of the pro-inflammatory IL-1β and MIP-2 was substantially decreased by sGAG. In contrast, the cytokines IL-6 and  TNF-α,which are reported to recruit progenitors, enhance vascularization and induce osteogenesis, were increased in the sGAG groups. These results suggest a positive effect of  sGAG on bone healing which is supported by previous in vivo and in vitro data. The investigation focused on the first stages of bone healing.

The introduction is well written , with adequate bibliographic references and stating the hypothesis of the study 

The methodology is complete, widely described, which would allow the study to be carried out by another research group. The statistical power of the sample size must be reflected Results are clearly described. The discussion is correct, adapting to the results obtained. The strengths must be included. A graphical representation of the interactions between cytokines could be interesting to add to understand the process

Author Response

The introduction is well written, with adequate bibliographic references and stating the hypothesis of the study. The methodology is complete, widely described, which would allow the study to be carried out by another research group. The statistical power of the sample size must be reflected. Results are clearly described. The discussion is correct, adapting to the results obtained. The strengths must be included. A graphical representation of the interactions between cytokines could be interesting to add to understand the process.

Thank you very much for your comments. The sample size was restricted to the animal number approved by the animal care committee.

We added a graphical overview of the effects of sulfated glycosaminoglycans on the cytokine concentrations observed in this study. As the identification and characterization of the interaction of various cytokines with each other was not within the scope of this manuscript, we believe that a graphical representation of that would not match the actual results achieved.

Reviewer 3 Report

In the manuscript, the authors mainly investigated the effect of GAG on the inflammation cytokines in the early stage of bone defect healing. The results indicated that the IL-1β and MIP-2 decreased with the treatment of GAG, however, the level of IL-6 and TNF-α increased. Several issues are required to be addressed.

(1) The author only observed the levels of inflammation cytokines in the early stage of bone healing, how could the authors get the conclusion that GAG treatment improve the bone healing? The author may need to observe and analyze the callus around the bone defect for a long-time study to further verify it.

(2) Did the author find any difference between PCL coated with collagen and chondroitin sulfate (CS) and PCL coated with collagen and hypersulfated hyaluronic acid (sHA3) groups?

Author Response

(1) The author only observed the levels of inflammation cytokines in the early stage of bone healing, how could the authors get the conclusion that GAG treatment improve the bone healing? The author may need to observe and analyze the callus around the bone defect for a long-time study to further verify it.

We have been working on the impact of sulfated glycosaminoglycans (sGAG) on bone regeneration for several years. During that time we observed improved bone healing in femoral defects in rats treated with sGAG and published several papers on that (Förster, Y., Schulze, S., Penk, A., Neuber, C., Möller, S., Hintze, V., ... & Rammelt, S. (2020). The influence of different artificial extracellular matrix implant coatings on the regeneration of a critical size femur defect in rats. Materials Science and Engineering: C, 116, 111157; Neuber, C., Schulze, S., Förster, Y., Hofheinz, F., Wodke, J., Möller, S., ... & Pietzsch, J. (2019). Biomaterials in repairing rat femoral defects: In vivo insights from small animal positron emission tomography/computed tomography (PET/CT) studies. Clinical Hemorheology and Microcirculation, 73(1), 177-194). Maybe this was not made clear in the manuscript. We changed the manuscript accordingly and now emphasize the correlation between the previous results and our new findings.

(2) Did the author find any difference between PCL coated with collagen and chondroitin sulfate (CS) and PCL coated with collagen and hypersulfated hyaluronic acid (sHA3) groups?

Thank you for this question. The concentrations of secreted IL-1β and IL-10 were indeed similar in animals that were treated with either collagen/CS or collagen/sHA3. However, the secretion of IL-6, MCP-1, MIP-2 and TNF-α varied in both, concentration and time when CS and sHA3 group were compared. The overall concentration of the tested cytokines did not differ significantly between the CS and sHA3 group. This is in line with our previous studies on the effects of sulfated glycosaminoglycans on bone regeneration (mentioned above) with no significant differences between CS and sHA3 group while the sGAG groups performed better than the positive control.

Round 2

Reviewer 1 Report

The revised version of the paper is now suitable for publication.

Reviewer 3 Report

Thank you for revising the manuscript. I do not have any more questions.